# Serine/Threonine Phosphatases in LTP: Two B or Not to Be the Protein Synthesis Blocker-Induced Impairment of Early Phase

**DOI:** 10.3390/ijms22094857

**Published:** 2021-05-04

**Authors:** Alexander V. Maltsev, Natalia V. Bal, Pavel M. Balaban

**Affiliations:** Institute of Higher Nervous Activity and Neurophysiology, Russian Academy of Sciences, 117485 Moscow, Russia; bal_nv@mail.ru (N.V.B.); pmbalaban@gmail.com (P.M.B.)

**Keywords:** long-term potentiation, anisomycin, cycloheximide, nitric oxide, protein phosphatases, calcineurin

## Abstract

Dephosphorylation of target proteins at serine/threonine residues is one of the most crucial mechanisms regulating their activity and, consequently, the cellular functions. The role of phosphatases in synaptic plasticity, especially in long-term depression or depotentiation, has been reported. We studied serine/threonine phosphatase activity during the protein synthesis blocker (PSB)-induced impairment of long-term potentiation (LTP). Established protein phosphatase 2B (PP2B, calcineurin) inhibitor cyclosporin A prevented the LTP early phase (E-LTP) decline produced by pretreatment of hippocampal slices with cycloheximide or anisomycin. For the first time, we directly measured serine/threonine phosphatase activity during E-LTP, and its significant increase in PSB-treated slices was demonstrated. Nitric oxide (NO) donor SNAP also heightened phosphatase activity in the same manner as PSB, and simultaneous application of anisomycin + SNAP had no synergistic effect. Direct measurement of the NO production in hippocampal slices by the NO-specific fluorescent probe DAF-FM revealed that PSBs strongly stimulate the NO concentration in all studied brain areas: CA1, CA3, and dentate gyrus (DG). Cyclosporin A fully abolished the PSB-induced NO production in the hippocampus, suggesting a close relationship between nNOS and PP2B activity. Surprisingly, cyclosporin A alone impaired short-term plasticity in CA1 by decreasing paired-pulse facilitation, which suggests bi-directionality of the influences of PP2B in the hippocampus. In conclusion, we proposed a minimal model of signaling events that occur during LTP induction in normal conditions and the PSB-treated slices.

## 1. Introduction

Long-term potentiation (LTP), long-term depression (LTD), and depotentiation are the key forms of synaptic plasticity at different brain regions: amygdala, hippocampus, cerebellum, etc. [1,2,3]. Modifications of excitability in neurons produced during LTP, LTD, or depotentiation are involved in processes underlying memory consolidation, memory reconsolidation, and learning in different animal species [4,5,6]. The LTP phenomenon, consisting of the strengthening of synaptic transmission between pre- and postsynaptic neurons due to high-frequency stimulation of afferent pathway, is the most studied form of plasticity [7,8,9]. It is conditionally divided into an early phase (E-LTP), which is protein synthesis-independent and lasts from tens of minutes to several hours, and a late phase (L-LTP), which is protein synthesis-dependent and persists for many hours [10]. Many molecular systems were shown to participate in the development of LTP: ligand-regulated or potential-dependent ion channels (NMDA and AMPA receptors; voltage-gated calcium, potassium, and sodium channels; etc.) [11,12,13], protein kinases (cAMP-dependent, PKA; cGMP-dependent, PKG; phosphatidylinositol-3-kinase, PI3K, Akt-kinase, Ca^2+^/calmodulin II kinase, CaMK II; etc.) [14,15,16,17], and phosphatases (PP1, PP2A, PP2B) [18,19,20]. L-LTP is to a certain extent supported by kinases activated during E-LTP, which are part of a family of extracellularly regulated kinases (ERKs). ERKs phosphorylate many cytoplasmic and nuclear proteins, including transcription factors: c-fos, jun, c-myc, CREB, etc. [21,22,23]. Ultimately, they change gene transcription and protein synthesis. It is interesting to note that the number of papers dedicated to the role of different protein kinase types in LTP development is an order of magnitude greater than the number of papers related to phosphatase influences. It is believed that phosphatases (mainly widespread in the brain are calcineurin and PP2B) are important for LTD or depotentiation [24,25]. However, some studies demonstrate the significance of phosphatases for LTP. Chronic stress in rats sharply decreased the PP2B protein levels in the dentate gyrus (DG) and E-LTP in CA1 hippocampal layer neurons [26]. Using a gene construct to express the PP2B inhibitor in a mouse brain, it was shown that transient reduction in calcineurin activity facilitates LTP in vitro and in vivo. These changes are accompanied by enhanced learning and strengthened short- and long-term memory in several hippocampal-dependent spatial and nonspatial tasks [27]. Furthermore, the PP2B blockers FK506 and rapamycin in conjunction with weak stimulation induced the NMDA-dependent LTP in rat hippocampal CA1 neurons [28]. Interesting experiments were carried out by Zeng et al., in which calcineurin activity in genetically modified mice was disrupted specifically in the adult forebrain. Strikingly, although performance was normal in hippocampus-dependent reference memory tasks, including contextual fear conditioning and the Morris water maze, the mutant mice demonstrated impairment in hippocampus-dependent working and episodic-like memory tasks, including the delayed matching-to-place task and the radial maze task. These data suggest a critical role for calcineurin in bidirectional synaptic plasticity [29]. Manipulations with overexpression of calcineurin revealed that PP2B has a role in the transition from short- to long-term memory [30].

Recently, we have shown that protein synthesis blockers (PSBs) cycloheximide and anisomycin suppressed both the early and late phases of LTP induced with a stimulation protocol of four 100 Hz bursts 5 min apart [31]. In these experiments, the increase in directly measured NO levels in PSB-treated hippocampal slices even in the absence of electric stimulation was observed. However, the mechanism of NO synthase (NOS) activation during protein synthesis blockade remains unclear [31,32]. It is well known that kinase–phosphatase switching significantly regulates the activity of NOS isoforms. For example, nNOS phosphorylation by CaMK II or protein kinase C (PKC) suppresses the enzyme activity [33,34,35], while phosphorylation by AMP-dependent kinase, protein kinase A, or Akt increases the NO production by nNOS [36,37,38,39]. Two principal residues in the nNOS complex are targets for modulation of its enzyme activity: ‘stimulating’ Serine-847 and ‘inhibiting’ Serine-1412. Calcineurin (PP2B) can directly dephosphorylate PKC-dependent or CaMK II-dependent phosphorylation of nNOS, enhancing its activity [40,41]. The alternative pathway for PP2B includes an indirect influence on the level of nNOS phosphorylation at Serine-847 that increases the NO production [42]. Recently, it was shown that PP2B inhibitor cyclosporin A decreased the nNOS-dependent neurite outgrowth in hippocampal neural progenitor cells [43]. Interestingly, a direct association of the C-terminal region of the transmembrane AMPA receptor regulatory proteins with PP2B was found, which regulates the AMPAR phosphorylation and trafficking into the plasmalemma [44]. It was shown that nNOS, AMPA receptors, and some kinases (e.g., CaMK II) and phosphatases are colocalized with the postsynaptic density proteins (PSDs) in glutamatergic neurons [45,46,47]. PSDs are known to be necessary for the formation of spines that are responsible for changes associated with the development of long-term potentiation and, as is commonly believed, ultimately participate in memory storage [48,49].

The aim of this work was to verify the contribution of calcineurin (PP2B) to nNOS activation, which occurs during blockade of protein synthesis in neurons. Data obtained show that the early-phase LTP deterioration observed in the PSB-pretreated hippocampal slices is at least partially mediated through an increase in phosphatases’ activity and PP2B-dependent activation of nNOS.

## 2. Results

### 2.1. PSBs Impair Both Pre- and Postsynaptic Plasticity

The ability of PSBs to depress L-LTP in different rat brain regions, namely spinal cord horn, mossy fibers, amygdala, dentate gyrus, and dendrites of CA1 layer of the hippocampus, is well known [50,51,52,53]. According to these data, anisomycin (25 μM) or cycloheximide (100 μM), the established inhibitors of translation, applied 40 min before tetanus impaired both the amplitude and slope coefficient of fEPSP in comparison to the control. The results of our experiments are presented in Figure 1A.

Notably, both blockers significantly impaired not only the protein synthesis-dependent L-LTP but also the E-LTP. The slope of fEPSP in presence of anisomycin immediately after LTP induction was reduced to 240.1 ± 37.6% vs. 334.8 ± 20.4% in control slices (*p* < 0.05, Figure 1A,B). Thirty minutes after LTP induction, the slope values were 184.3 ± 18.6% and 284.4 ± 18.5% for anisomycin-treated and control slices, respectively (*p* < 0.05). The effect of cycloheximide (100 μM) was similar to that of anisomycin: impaired slope of fEPSP was 220.3 ± 21.8% at 0–3 min after LTP induction (*p* < 0.05) and 194.5 ± 15.6% 30 min after LTP induction (for both, *p* < 0.05 vs. control at the corresponding time). Figure 1B shows the first response immediately after LTP induction in control (green circles), cycloheximide-treated (dark red circles), and anisomycin-treated (red circles) slices, demonstrating an obvious PSB-induced E-LTP deterioration.

To test for any modification of the presynaptic function, we investigated the difference in the paired-pulse facilitation (PPF) in the CA1 neurons when two identical stimuli were delivered at six different interstimulus intervals (ISI) to the Schaffer collaterals. PPF is the presynaptic form of the short-term plasticity in which the synaptic response to the second paired stimulus in rapid succession is increased due to residual Ca^2+^ in the presynaptic terminal from the first stimulus. Therefore, the extent of facilitation is increased with shorter ISI. It is curious that 25 µM anisomycin decreased PPF at ISI of 50 and 100 ms after 40 min incubation of the pre-tetanic slices (for 50 ms, 1.38 ± 0.04 vs. 1.58 ± 0.06; for 100 ms, 1.29 ± 0.03 vs. 1.42 ± 0.05; for both, *p* < 0.05; Figure 1C, top panel). This tendency was present for E-LTP at 30 min after LTP induction. At this time, the PPF in anisomycin-treated slices was significantly suppressed at 30, 50, and 100 ms ISI (for 30 ms, 1.39 ± 0.07 vs. 1.56 ± 0.08; for 50 ms, 1.31 ± 0.06 vs. 1.45 ± 0.07; for 100 ms, 1.26 ± 0.06 vs. 1.34 ± 0.05; for all, *p* < 0.05; Figure 1C, bottom panel).

### 2.2. nNOS Is Involved in PSB-Induced Impairment of the E-LTP

3-Br-7NI, a specific inhibitor of neuronal NOS, applied to hippocampal slices 40 min before LTP induction in a concentration of 5 µM, significantly decreased the L-LTP (to 173.8 ± 4.5% vs. 212.1 ± 8.1%, *p* < 0.05, Figure 2A,B, right panel), without significant influence on the E-LTP (301.9 ± 19.9% vs. 316.1 ± 19.7%, *p* > 0.05, Figure 2B, left panel). Anisomycin application, as expected, significantly decreased both phases of LTP (Figure 2 A,B).

Simultaneous addition of 3-Br-7NI (5 µM) + anisomycin (25 µM) prevented the ANI-induced LTP impairment in both the early and late phases (Figure 2A,B). Slope values were 306.4 ± 19.6% and 209.9 ± 12.2% for E-LTP and L-LTP, respectively (for both, *p* > 0.05 vs. corresponding slopes in control slices). The results obtained suggest a significant contribution of nNOS to the PSB-related modifications of synaptic plasticity in the hippocampus.

### 2.3. PP2B Is Involved in the PSB-Induced Impairment of E-LTP in CA1 Neurons

An established PP2B inhibitor, cyclosporin A, applied to hippocampal slices 40 min before LTP induction in a concentration of 5 µM did not produce any significant changes in synaptic plasticity during either stage of LTP development (283.2 ± 18.1% vs. 316.1 ± 19.7% and 213.4 ± 9.5% vs. 212.1 ± 8.1% for E-LTP and L-LTP, respectively; for both, *p* > 0.05, Figure 3A,B).

Application of cyclosporin A under PSB prevented the PSB-induced impairment of E-LTP (286.5 ± 24.9% vs. 222.6 ± 32.9% and 267.3 ± 22.6% vs. 220.3 ± 21.9% for anisomycin and cycloheximide, respectively; for both *p* < 0.05; Figure 3B,D, left panels). It should be noted, that cyclosporin A + PSB elicits deterioration in the L-LTP in the same manner as PSB alone (173.8 ± 4.4% vs. 155.3 ± 12.0% and 141.9 ± 12.6% vs. 123.2 ± 11.4% for cycloheximide and anisomycin, respectively; for both *p* > 0.05; Figure 3B,D, right panels). These results indicate that PP2B is selectively responsible for the PSB-induced impairment of E-LTP but not L-LTP.

### 2.4. PP2B Blocker Prevents PSB-Induced NO Production in the Hippocampus

In the next series of experiments, we directly measured the NO production in PSB-treated and control slices. In non-stimulated hippocampal slices, 25 µM anisomycin or 100 µM cycloheximide significantly increased the NO-sensitive fluorescence of the DAF-FM probe (Figure 4A,C).

In the CA1 dendrites (stratum lacunosum-moleculare, SLM), NO synthesis was increased to 125.3 ± 3.5% and 135.2 ± 2.4% for anisomycin and cycloheximide, respectively. In the CA3 dendrites, NO production was increased to 115.8 ± 2.1% and 121.5 ± 1.9% for anisomycin and cycloheximide, respectively. In the DG dendrites, NO synthesis was increased to 132.4 ± 3.7% and 133.2 ± 2.9% for anisomycin and cycloheximide, respectively (Figure 4C). Intriguingly, cyclosporin A almost fully abrogated the DAF-FM fluorescence increase in PSB-treated slices in all studied regions (CA1, CA3, DG) (Figure 4B,C). Thus, in the anisomycin + cyclosporin A-treated slices, the DAF-FM fluorescence values were 101.3 ± 1.3%, 102.6 ± 2.2%, and 103.1 ± 1.8% for CA1, CA3, and DG, respectively. This tendency was also observed after tetanization of the PSB-treated slices, where 5 µM cyclosporin A effectively prevented the cycloheximide- or anisomycin-dependent NO production in CA1, CA3, and DG (Figure 5).

For cyclosporin A (5 µM) + anisomycin (25 µM)-pretreated slices, the changes in DAF-FM fluorescence were 103.4 ± 2.3%, 102.3 ± 1.7%, 102.6 ± 2.8% for CA1, CA3, and DG, respectively; for all, *p* ˂ 0.05 relative to respective values under tetanus, CHM (100 µM) or ANI (25 µM) (Figure 5B,C). Similar data were observed for cycloheximide (Figure 4C and Figure 5C, Appendix A). Taken together, data obtained suggest that PP2B mediates the PSB-induced NO production and E-LTP impairment in the hippocampus.

### 2.5. Influence of PP2B on Short-Term Plasticity

Cyclosporin A (5 µM) applied to hippocampal slices for 40 min decreased the PPF at 30 and 50 ms interstimulus intervals (1.59 ± 0.07 vs. 1.83 ± 0.12 and 1.42 ± 0.04 vs. 1.53 ± 0.05 for 30 and 50 ms, respectively, Figure 6A,B). Surprisingly, cyclosporin A did not restore the impaired PPF induced by 25 µM anisomycin application before LTP induction (e.g., for 50 ms, 1.43 ± 0.04 vs. 1.38 ± 0.04; for 100 ms, 1.33 ± 0.03 vs. 1.29 ± 0.03; for both, *p* > 0.05) or during E-LTP (at 30 min after tetanus: 1.37 ± 0.03 vs. 1.31 ± 0.06 and 1.30 ± 0.02 vs. 1.29 ± 0.03 for 50 and 100 ms ISI, respectively; for both, *p* > 0.05), despite the fact that it was effective in the preservation of E-LTP in the presence of anisomycin (Figure 6A,B).

Cyclosporin A influence on PPF is apparently associated with a moderate increase in both the amplitude and slope of the first stimulus rather than with a decrease in these parameters of the second stimulus (Figure 6C). After 20 min pretreatment of slices by 5 μM cyclosporin A, the amplitudes of first responses were increased to 115.6 ± 5.9% (Figure 6C). Thus, PP2B showed bi-directional effects: it mediates the PSB-induced impairment of E-LTP and at the same time is needed for normal functioning of short-term plasticity.

### 2.6. Direct Measurements of Serine/Threonine Phosphatase Activity in the Hippocampus

In our experiments, it was observed that 25 µM anisomycin addition to hippocampal slices led to the increase in phosphatase activity to 150.4 ± 11.8% of control values (*n* = 4, *p* ˂ 0.05, Figure 7A). This increase was fully blocked in the presence of the nNOS specific blocker 3-Br-7NI, 5 µM (up to 100.2 ± 9.5%, *n* = 4, Figure 7A). Furthermore, the NO donor SNAP (20 µM) increased the phosphatase activity in the hippocampal slices in the same manner as PSB (to 159.0 ± 20.3%, *n* = 4, *p* ˂ 0.05). It is curious that simultaneous pretreatment of slices by SNAP (20 µM) + anisomycin (25 µM) did not reveal any further increase in serine/threonine phosphatase activity (158.1 ± 21.4%, *n* = 4, *p* ˂ 0.05, Figure 7A), suggesting that PSB- and NO-dependent signaling pathways are convergent with respect to PP2B activity.

Tetanic stimulation of slices sharply decreased serine/threonine phosphatase activity to 71.5 ± 5.6% (*n* = 4, *p* ˂ 0.05, Figure 7B) of control values. It was observed that in anisomycin (25 µM)-treated slices 30 min after LTP induction, the phosphatase activity was somewhat less than that of the control (81.8 ± 6.0%, *p* ˂ 0.05, *n* = 4, Figure 7B). Furthermore, surprisingly, in non-tetanized slices, cyclosporin A (5 µM) inverted the influence of anisomycin (25 µM) on the phosphatase activity (to 81.9 ± 4.8%, *p* ˂ 0.05, *n* = 4, Figure 7B). It is intriguing that, with respect to the influence of a PSB on the serine/threonine phosphatase activity, tetanus and PP2B blocker act similarly, because tetanus + anisomycin (25 µM) and cyclosporin A (5 µM) + anisomycin (25 µM) led to the inversion of anisomycin-induced phosphatase activity.

## 3. Discussion

The addition or removal of negatively charged phosphate residues at a targeted protein leads to conformational changes, causing a change in its functional activity. This is one of the elemental and evolutionarily conservative mechanisms found in both prokaryotes and eukaryotes. Both the kinases responsible for the attachment of phosphate to the protein and the phosphatases that remove it are ubiquitously distributed in the body [54,55,56,57]. Almost the entire spectrum of known kinases and phosphatases is expressed in the brain [57,58,59]. The involvement of serine/threonine phosphatases in LTP maintenance has been shown for some brain structures. For example, stimulating the Purkinje fibers in the cerebellum elicited a decrease in the fEPSP amplitude under the phosphatase 1 and 2A blockers [60]. A phosphatase 2B inhibitor, FK 506, suppressed the induction of LTP, which is sensitive to the blockade of voltage-gated Ca^2+^ channels in the hippocampus or cerebellum [61,62]. At the same time, the mechanisms for the reduction in both the fEPSP amplitude and speed during modulation of phosphatase activity are still unclear. In particular, there are no data on the relationship of the serine/threonine phosphatase activity to the efficiency of neuronal NO synthase and nitric oxide production. NO, being a retrograde neurotransmitter, modulates both presynaptic and postsynaptic plasticity and is involved in such key processes as reconsolidation, memory storage, and learning [63,64,65,66,67].

In this study, we focused on the role of phosphatase 2B (calcineurin, PP2B) because this enzyme is the most expressed phosphatase in hippocampal tissues and appears to play an important role [68,69]. PP2B is a serine/threonine phosphatase, activated by the Ca^2+^/calmodulin complex, involved in the signaling pathways required for gene expression, as well as in biological responses to external stimuli in many organisms and in various types of cells [70,71,72]. It was shown that PP2B directly controls NFAT signaling; NFAT signaling is involved in neuron outgrowth, which is important for the plasticity and hippocampal functioning [71,73]. By binding to the regulatory subunit of NFAT proteins, PP2B dephosphorylates it, initiating NFAT translocation into the nucleus, binding to DNA, and activation of gene transcription [74]. NFAT counteracts with neurotrophic signal transduction to control the axon outgrowth in several types of nerve cells [75,76]. Furthermore, activation of calcineurin/NFAT signaling facilitates the formation of new synapses and helps to build neural circuits in the brain [75,77]. NFAT is a known important player in both the developing and adult nervous systems [75,78].

PP2B is a heterodimer consisting of catalytic (59 kDa) and Ca^2+^-binding regulatory (19 kDa) subunit [79,80]. Moreover, the catalytic subunit of the enzyme, in addition to the catalytic domain itself, includes a CaM-binding modulatory site and the C-terminal autoinhibitory domain, the removal of which leads to permanent activation of PP2B [79,81]. The autoinhibitory domain forms a loop closing the active site of the catalytic domain, thus preventing its interaction with NFAT in resting cells [82]. Interestingly, PP2B activity in the hippocampus and anterior cortex is upregulated in patients with Alzheimer’s disease [83,84].

It was shown in central synapses that PP2B expression is revealed in both the pre- and postsynaptic structures [85,86]. Accordingly, there is evidence of a possible involvement of PP2B in the regulation of synaptic transmission at both the post- and presynaptic levels [87,88,89]. In central synapses, postsynaptic PP2B was demonstrated to participate in regulation of long-term depression (LTD), activity of L-type Ca^2+^-channels, and degradation of postsynaptic NMDA receptors [88,90,91,92]. At the presynaptic level, PP2B was found to be involved in the modulation of vesicle endocytosis during long-term high-frequency synaptic activity [93]. In a primary culture of hippocampal neurons, PP2B facilitated the release of neurotransmitters from the reserve pool during electrical stimulation [94]. These data correlate well with those obtained in our study. In hippocampal slices pretreated by the PP2B inhibitor cyclosporin A, the paired-pulse facilitation ratio (PPFR) was reduced, showing an attenuation in short-term plasticity (Figure 6). Analysis of PPFR decrement shows that it is associated with a moderate increase in amplitude of the first response rather than a decrease in amplitude of the response to the second stimulus. Our results suggest that PP2B in presynapse facilitates only the first response with some refractoriness for subsequent ones. Whether PP2B ceases to play a significant role in the second and/or subsequent response(s) or its action is masked by some kind of negative feedback remains to be clarified. Nevertheless, the contribution of PP2B to the short-term plasticity in the CA1 layer of the hippocampus is obvious and may be physiologically significant for some types of memory and learning.

In the present study, we also investigated the relationship between NO-dependent suppression of LTP caused by the application of the protein synthesis blockers (PSBs) and PP2B. Both the nNOS specific inhibitor, 3-Br-7NI (5 µM), and cyclosporin A (5 µM) prevented the decrease in LTP in the early phase, which occurs during the first minutes after tetanus induction in the anisomycin (25 µM)- or cycloheximide (100 µM)-pretreated slices. Using a highly specific NO-sensitive DAF-FM probe, we observed that cyclosporin A (5 µM) abolishes the PSB-induced NO synthesis in all layers of the hippocampus, including CA1, CA3, and dentate gyrus (DG). At the same time, in hippocampal slices, application of cycloheximide or anisomycin for 20–40 min induced a significant increase in NO production even in the absence of tetanic stimulation. Obviously, a special role of PP2B in the PSB-induced LTP impairment can be suggested, so far as the common PP1/PP2A inhibitors, okadaic acid and calyculin A, did not prevent the E-LTP decline in the PSB-treated slices (Appendix A).

It is curious to note the multidirectional effects of PP2B on synaptic plasticity: the preservation of the early LTP phase during the PSB treatment in the presence of cyclosporin A suggests the ‘impairing’ role of PP2B in E-LTP, while a decrease in PPFR by cyclosporin A suggests the facilitating role of PP2B in short-term plasticity. Apparently, these bidirectional actions occur independently, because prevention of the impairment of E-LTP by anisomycin with cyclosporin A at the same time does not restore the PPFR in pre-tetanically anisomycin-treated slices or at 30 min after LTP induction. The PPFR is very often associated with facilitation of calcium-dependent release of neurotransmitters from the presynapse, and it is logical to speculate that these actions can be spatially limited: facilitation of short-term plasticity by PP2B is implemented in presynaptic structures, and PSB-induced suppression of early phase is apparently associated with postsynaptic mechanisms.

Direct measurements of serine/threonine phosphatase activity in hippocampal slices in our experiments showed that the PSB applications lead to its substantial increase, which is inhibited in the presence of cyclosporin A. In addition, tetanic stimulation sharply suppressed the phosphatase activity, which was partially restored in the PSB-treated slices. Application of the NO donor SNAP mimicked the PSB effects, while its simultaneous application with anisomycin did not potentiate the phosphatase activity, which implies either the saturation of NO-mediated influences with respect to serine/threonine phosphatase activity or the convergency of the NO–PP2B interactions. Based on the obtained data, a simplified model of events occurring in the PSB-treated tetanized hippocampal slices is proposed (Figure 8). It not only suggests the significance of PSB influences associated with inhibition of translation processes but also stresses the role of the adaptive neurochemical events detected in this study. In this model (Figure 8), we used our results to try to describe possibilities for regulation of the kinase–phosphatase balance: (A) At quiescent state, the kinase–phosphatase balance in cells is shifted towards dephosphorylation processes, because most proteins, including ion channels, enzymes, and transcription factors, normally have dephosphorylated status. During the induction of tetanus, there is a sharp switch in the balance towards kinase activity, as multiple protein kinase isoforms are activated in both the Ca^2+^-dependent and -independent manners. (B) For the E-LTP development, a new kinase–phosphatase balance is established, which is responsible for increased synaptic transmission in CA1. Pretreatment of the slices by the PSB leads to the NO-dependent partial reversal of the tetanus-induced kinase–phosphatase balance that changes back to an increase in phosphatase activity (C), which, in its turn, leads to a partial suppression of the E-LTP (in both the amplitude and slope of fEPSP in the CA1 area of the hippocampus). Still, blockade of PP2B by cyclosporine A, applied simultaneously with the PSB, leads to switching of the PSB-induced reversal of balance to the state similar to that of the single tetanus, at least for the early phase of LTP (D).

Unfortunately, measurements of serine/threonine phosphatase activity cannot completely exclude the possible contribution of other phosphatases that also can be modulated by PP2B. Thus, PP2B is able to influence the activity of PP1 due to the dephosphorylation of its intracellular inhibitor protein (I-1). In the phosphorylated state, I-1 inhibits PP1, but its dephosphorylation by PP2B prevents this negative effect of I-1, leading to disinhibition of PP1 activity [95]. Further studies in this direction will help to understand better the picture of events underlying the LTP phenomenon. Regulation of kinase–phosphatase switching may be useful for clinical practice, as a number of neurodegenerative diseases are known to be associated with changes in the phosphorylation/dephosphorylation status of some neuronal proteins. Compensation of shifts in the kinase–phosphatase balance may be useful for the treatment of such pathological states.

## 4. Conclusions

A combination of inhibitory analysis and direct measurements of both NO production and serine/threonine phosphatase activity in the hippocampus allowed a deeper understanding of the processes underlying the PSB-induced impairment of LTP. PSBs suppressed the early phase of LTP development and hippocampal short-term plasticity regardless of their effects on translation processes. PSB action depended on activation of PP2B and neuronal NOS and was abolished by their blockade with specific pharmacological tools. The results obtained in our experiments also suggest the involvement of serine/threonine phosphatases in pathological states in the hippocampus, as significant changes in the expression of serine/threonine phosphatase activity were observed in a number of models of nerve tissue damage, such as spinal cord injury, ischemia/hypoxia, and head injury [86,96,97], as well as in some neurogenerative diseases [83,84,98].

## 5. Materials and Methods

### 5.1. Slice Preparation

All experiments followed the European Convention for the Protection of Vertebrate Animals used for Experimental and other Scientific Purposes 1986 86/609/EEC and the institutional requirements for the care and use of laboratory animals. Male Wistar rats (6–8 weeks old) were anesthetized by sevoflurane and decapitated. Brains were quickly submerged in ice-cold dissection solution (124 mM NaCl, 3 mM KCl, 1.25 mM NaH_2_PO_4_, 26 mM NaHCO_3_, 0.5 mM CaCl_2_, 7 mM MgCl_2_, and 10 mM D-glucose; pH equilibrated with 95% O_2_ and 5% CO_2_). Transverse (parasagittal) hippocampal slices (400 µm thick) were prepared using a vibratome Leica VT1000S (Leica Biosystems, Wetzlar, Germany) and immediately transferred to a recording (ACSF) solution (composition as above, except the CaCl_2_ and MgCl_2_ concentrations were adjusted to 2.5 and 1.3 mM, respectively). Slices were incubated at 34 °C in a water bath for 40 min and then kept at room temperature in aerated ACSF.

### 5.2. Electrophysiology

During the experiments, slices were perfused by a continuously flowing (appr. 4 mL/min) recording solution at 33–34 °C. Electrophysiological recordings were carried out using a SliceMaster system (Scientifica, Uckfield, UK). Field excitatory postsynaptic potentials (fEPSP) were recorded from stratum radiatum in area CA1 using glass microelectrodes (1–2 MΩ) filled with the recording solution. Baseline synaptic responses were evoked by paired-pulse stimulation with 50 ms interval of the Schaffer collaterals at 0.033 Hz with a bipolar electrode. Test stimulation intensity was adjusted to evoke fEPSP with amplitude 50% of maximal and was kept constant throughout the experiment. Long-term potentiation was induced with four 100 Hz bursts spaced 5 min apart as in [99]. The data were recorded and analyzed using Spike2 (Cambridge Electronic Design Limited, Cambridge, UK) and SigmaPlot 11.0 (Systat Software Inc, Chicago, IL, USA). For statistical analysis, the first 3 min after tetanization (0–3 min, early LTP) and the last 3 min (178–180 min after LTP induction, late LTP) were used. Paired pulse facilitation (PPF) ratio was calculated as follows: PPF = (S_2_EPSP/S_1_EPSP), where S_1_EPSP and S_2_EPSP are the slopes of EPSP in response to the first and the second stimuli with different intervals, respectively. PPF measures were carried out just before and after LTP induction.

### 5.3. NO Imaging in Hippocampal Slices

Parasagittal slices (400 μm thick) after an hour of incubation in the 95% O_2_/5% CO_2_ perfused ACSF at 34 °C were transferred into the light-shielded camera containing aerated ACSF with bath-added 5 μM of the NO-sensitive dye DAF-FM diacetate and 0.05% Pluronic F-127. After an hour of incubation for the dye loading, the slices were twice washed in ACSF. Further, the slices were placed in the experimental chamber and exposed to the drug treatment and/or tetanus. Control slices were maintained in aerated ACSF without any influences. At the end of the experiment, slices were fixed in the 4% formaldehyde in PBS solution (concentrations in mM: 137 NaCl, 2.7 KCl, 10 Na_2_HPO_4_, 1.8 KH_2_PO_4_, pH 7.4) for 20 min, and then the formaldehyde was entirely replaced with PBS. Fixation and storage of slices occurred in the light-shielded plates. Further, the slices were imaged using a fluorescence microscope Keyence BZ-9000 (Keyence Corp., Osaka, Japan) equipped with mercury lamp, objective lens (X2, X10, X20), excitation filter (450–490), dichroic mirror, and emission filter (520–540). Optical images were recorded on CCD camera and further analyzed by BZ-II Viewer and Analyzer (Keyence Corp., Osaka, Japan) and ImageJ (NIH). For semiquantitative estimation of the NO production, the ratio of digitized dye signals in the layer dendrites (CA1, CA3, DG) to the DAF-FM fluorescence in the region of neuron soma in the same layer was used. For the precision of normalization, the area from which the average signal was calculated (region of interest, ROI) was unchanged both in the dendrites and soma layers. Hippocampal layer locations were ascertained under transmitted light.

### 5.4. Phosphatase Assay

Serine/threonine phosphatase activity was determined by using a nonradioactive molybdate dye-based phosphatase assay kit Promega (Promega, Madison, WI, USA) according to the manufacturer’s recommendations, as reported in [100]. Twenty minutes after tetanization of control or drug-treated slices (total time of the tested drug exposure to hippocampal slices in aerated ACSF was 40 min), slices were harvested and homogenized at 3 slices per 1 mL of buffer containing the following: sucrose, 250 mM; β-mercaptoethanol, 15 mM; EDTA, 0.1 mM; phenylmethylsulfonyl fluoride, 0.1 mM; TRIS-HCl, 50 mM (pH 7.4). Free phosphate was removed from the lysate supernatants using a Sephadex G-25 resin spin column. Phosphatase reactions were assessed in 50 μL samples at 37 °C in buffer containing 50 mM imidazole, 0.2 mM EGTA, 0.02% β-mercaptoethanol, and 0.1 mg/mL bovine serum albumin (pH 7.2) based on the dephosphorylating rate of the synthetic 754 Da phosphopeptide RRA[pT]VA, a substrate for PP2B. The specificity of the phosphatase reaction was tested by cyclosporin A (5 μM), a serine/threonine phosphatase 2B inhibitor. The reaction was stopped by the addition of the dye–additive mixture (50 μL), and samples were incubated to color development for 15 min at room temperature. Absorbance was measured at 600 nm using an Infinite 200 Pro plate reader (Tecan, Salzburg, Austria). For normalization of phosphatase activity, we measured total protein amounts in hippocampal lysates using BCA Protein Assay Kit (Thermofisher, Waltham, MA, USA) according to the manufacturer’s recommendations. Protein-dependent reduction of Cu^2+^ to Cu^+^ in an alkaline medium followed by the bicinchoninic acid-sensitive detection of Cu^+^ ions was determined in hippocampal samples using bovine serum albumin (Amresco Inc., Solon, OH, USA) as a calibration standard (125–2000 µg/mL). Fifty microliter samples were incubated in the well of microplates at 37 °C for 30 min to color development and measured by Infinite 200 Pro plate reader (Tecan, Salzburg, Austria) at 562 nm.

### 5.5. Drugs

Salts for solutions, DMSO, cycloheximide, cyclosporin A, 3-Br-7NI, l-arginine, and NMDA were purchased from Sigma-Aldrich (St. Louis, MO, USA). SNAP was purchased from Tocris (Bristol, UK). Anisomycin was from Enzo Life Science (Enzo Biochem, New York, NY, USA). Fluorescent dye DAF-FM diacetate and DAPI were purchased from Molecular Probes (New York, NY, USA).

### 5.6. Statistical Analysis

Results are presented as mean ± standard error (S.E.M.) of *n* slices from at least three different animals (*n* = 3–12, depending on the experimental series). All statistical tests were performed using SigmaPlot 11.0 (Systat Software Inc., San Jose, CA, USA) or using SPSS Statistics software (version 21, IBM Corp., Armonk, NY, USA). The significance of changes in the fEPSP characteristics, the DAF-FM fluorescence, and the phosphatase activity was tested by ANOVA with the Bonferroni test. Phosphatase reaction values are expressed as percentage of control measurements (taken as 100%) and normalized to the total protein amounts in the sample. All tests used were two-sided; *p* < 0.05 was predetermined as defining statistically significant differences and in figures is denoted by * or ^#^ for multiple comparisons.

## Figures and Tables

**Figure 1 ijms-22-04857-f001:**
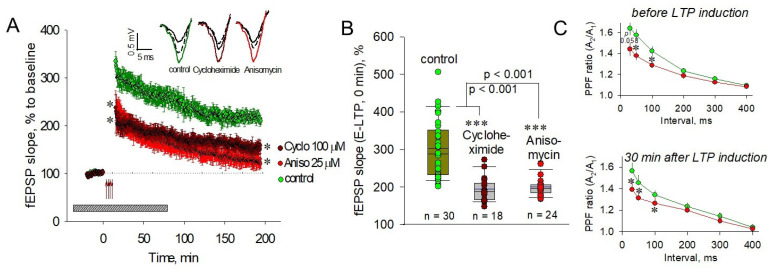
PSBs suppress both LTP and short-term plasticity in CA1 hippocampus. (**A**) Application of 100 µM cycloheximide (dark red circles) or 25 µM anisomycin (red circles) impairs both the early and late phases of LTP development in comparison to control tetanized slices (green circles). Top panels represent typical experimental recordings of field excitatory postsynaptic potentials (fEPSP) in the CA1 hippocampus for pre-tetanic responses (black lines), immediately after tetanic induction (green line—control, dark red line—Cycloheximide (100 µM)-treated, red line—Anisomycin (25 µM)-treated slices); and in the end of experiments (dashed lines). Here and below: the duration of drug infusion is shown as a gray rectangle at the bottom, and the dotted line shows the pre-tetanic level. The time of the beginning of the tetanus is marked by red arrows (0 min at the scale). (**B**) Bars containing first values of the fEPSP slope immediately after the end of tetanus in control (green circles) and in the presence of 100 µM cycloheximide (dark red circles) or 25 µM anisomycin (red circles). (**C**) In 25 µM anisomycin-pretreated slices, paired-pulse stimulation reveals the decreased PPFR before tetanus (40 min application, top panel) and during E-LTP, 30 min after the end of tetanic stimulation (bottom panel). * *p* ˂ 0.05 vs. the control, *** *p* ˂ 0.001.

**Figure 2 ijms-22-04857-f002:**
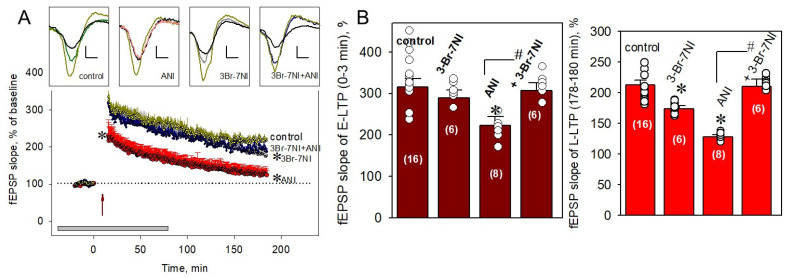
Inhibitor of neuronal NO synthase rescues the PSB-induced impairment of LTP. (**A**) Application of 5 µM 3-Br-7NI, the specific nNOS inhibitor, rescued both the early and late phases of LTP in the presence of 25 µM anisomycin (blue circles). 3-Br-7NI alone did not influence the E-LTP but slightly suppressed L-LTP development (gray circles). Red and dark yellow circles correspond to anisomycin (25 µM)-treated and control slices, respectively. The time of the beginning of the tetanus is marked by red arrow (with four 100 Hz bursts spaced 5 min apart, 0 min at the scale). The duration of drug applications is indicated by a gray rectangle at the bottom of graph. Top panels represent typical experimental recordings of field excitatory postsynaptic potentials (fEPSPs) in the CA1 hippocampus for pre-tetanic responses (black lines), immediately after tetanus induction (dark yellow lines), and in the end of experiments for control (green line), anisomycin (25 µM)-treated (red line), 3-Br-7NI (5 µM)-treated (gray line), 3-Br-7-NI (5µM) + Anisomycin (25 µM)-treated (blue line) slices. Scale bars are 0.5 mV and 5 ms for the Y-axis and X-axis, respectively. ANI—anisomycin. (**B**) Averaged data for E-LTP (left panel) and L-LTP (right panel) during the 3-Br-7NI (5 µM) or 3-Br-7NI (5 µM) + anisomycin (25 µM) applications. ANI, anisomycin. Number of experiments is indicated in parentheses. * *p* ˂ 0.05 vs. the control, # *p* ˂ 0.05 for comparison of groups between themselves.

**Figure 3 ijms-22-04857-f003:**
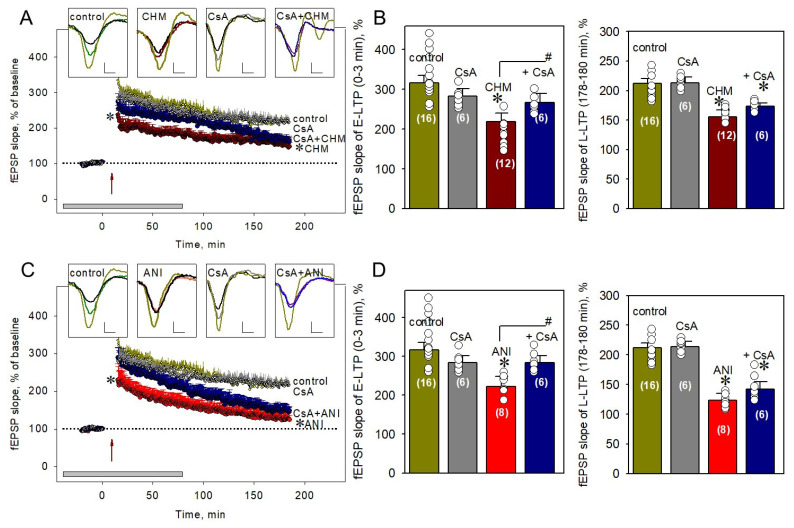
PP2B blocker prevents the PSB-induced impairment of E-LTP but not L-LTP. (**A**) Application of 5 µM cyclosporin A alone did not change the LTP kinetics (gray circles) but restored the 100 µM cycloheximide-induced decline of E-LTP (blue circles). Red and dark yellow circles correspond to cycloheximide (100 µM)-treated and control slices, respectively. The time of the beginning of the tetanus is marked by red arrow (with four 100 Hz bursts spaced 5 min apart, 0 min at the scale). The duration of drug applications is indicated by a gray rectangle at the bottom of graph. Top panels represent typical experimental recordings of field excitatory postsynaptic potentials (fEPSPs) in the CA1 hippocampus for pre-tetanic responses (black lines), immediately after tetanus induction (dark yellow lines), and in the end of experiments for control (green line), cycloheximide (100 µM)-treated (red line), Cyclosporin A (5 µM)-treated (gray line), Cyclosporin A (5µM) + Cycloheximide (100 µM)-treated (blue line) slices. Scale bars are 0.5 mV and 5 ms for the Y-axis and X-axis, respectively. CHM—Cycloheximide, CsA—Cyclosporin A. (**B**) Averaged data for (A), number of experiments is indicated in parentheses. (**C**) Cyclosporin A-mediated attenuation of 25 µM anisomycin influence on LTP (blue circles). Dark yellow, gray, and red circles correspond to control, Cyclosporin A (5 µM)-treated, and Anisomycin (25 µM)-treated slices. (**D**) Averaged data for (C). CsA, cyclosporin A; ANI, anisomycin. Number of experiments is indicated in parentheses. * *p* ˂ 0.05 vs. the control, # *p* ˂ 0.05 for comparison of groups between themselves.

**Figure 4 ijms-22-04857-f004:**
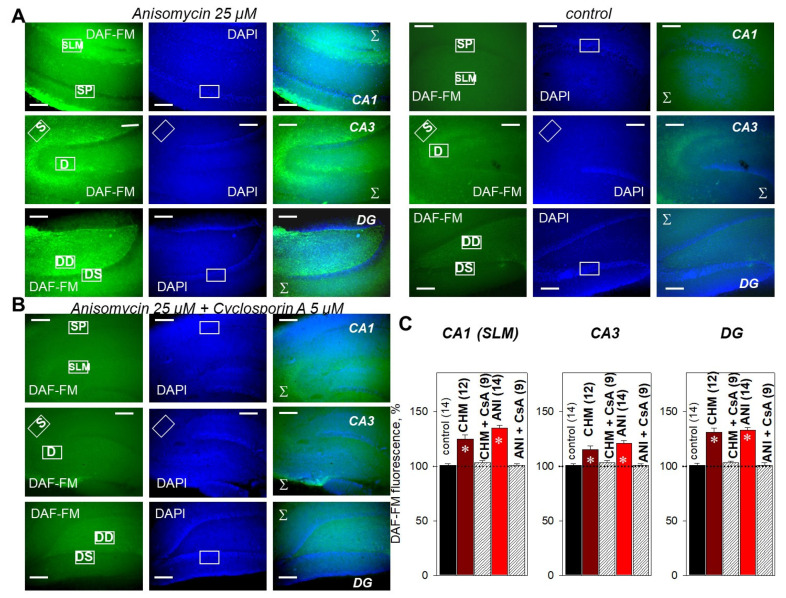
Fluorescent imaging of NO production in hippocampal slices. (**A**) NO production in PSB-treated (Anisomycin, 25 µM, three left panels), and control slices (three right panels) measured in the CA1, CA3, and dentate gyrus (DG). NO production was estimated by the intensity of fluorescence of the NO-specific DAF-FM probe (green channels, left panels). Blue staining shows cell nuclei (DAPI, middle panels). Right panels display channel overlays (Σ). (**B**) Cyclosporin A (5 µM) prevented the NO synthesis induced by 25 µM anisomycin in all studied regions of hippocampus. Scale bars 500 µm. SP, stratum pyramidale; SLM, stratum lacunosum-moleculare; S, soma layer in CA3; D, dendrite layer in CA3; DS, soma layer in DG; DD, dendrite layer in DG. (**C**)—averaged data for DAF-FM fluorescence in CA1, CA3, and DG. CHM—Cycloheximide (100 µM), CsA—Cyclosporin A (5µM), ANI—Anisomycin (25 µM). For the precision of normalization, the area from which the average signal was calculated (region of interest, ROI) was unchanged both in the dendrite and soma layers. Hippocampal layer locations were ascertained under transmitted light and DAPI channels. Number of independent experiments is indicated in parentheses. * *p* ˂ 0.05 vs. the control.

**Figure 5 ijms-22-04857-f005:**
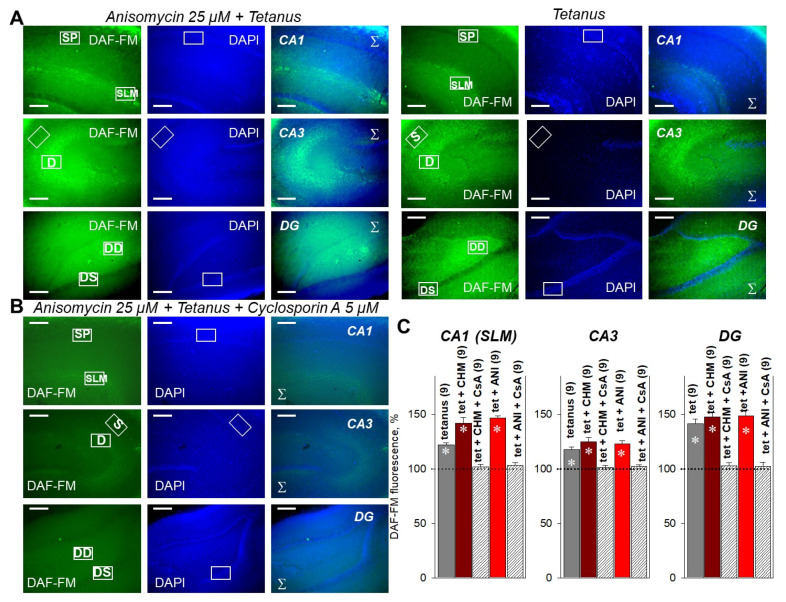
Fluorescent imaging of NO production in hippocampal slices after the induction of tetanus. (**A**) NO production after the tetanus induction (30 min) in PSB-treated (Anisomycin 25 µM, three left panels) and control slices (three right panels) measured in the CA1, CA3, and dentate gyrus (DG). The NO production was estimated by fluorescence of NO-specific DAF-FM probe (green channels, left panels). Blue staining shows cell nuclei (DAPI, middle panels). Right panels display channel overlays (Σ). (**B**) Cyclosporin A (5 µM) prevented the NO synthesis induced by 25 µM anisomycin + tetanus in all studied regions of the hippocampus. Scale bars 500 µm. SP, stratum pyramidale; SLM, stratum lacunosum-moleculare; S, soma layer in CA3; D, dendrite layer in CA3; DS, soma layer in DG; DD, dendrite layer in DG. (**C**) Averaged data for DAF-FM fluorescence in CA1, CA3, and DG. CHM—Cycloheximide (100 µM), CsA—Cyclosporin A (5 µM), ANI—Anisomycin (25 µM). For the precision of normalization, the area from which the average signal was calculated (region of interest, ROI) was unchanged both in the dendrite and soma layers. Hippocampal layer locations were ascertained under transmitted light and DAPI channels. Number of independent experiments is indicated in parentheses. * *p* ˂ 0.05 vs. the control.

**Figure 6 ijms-22-04857-f006:**
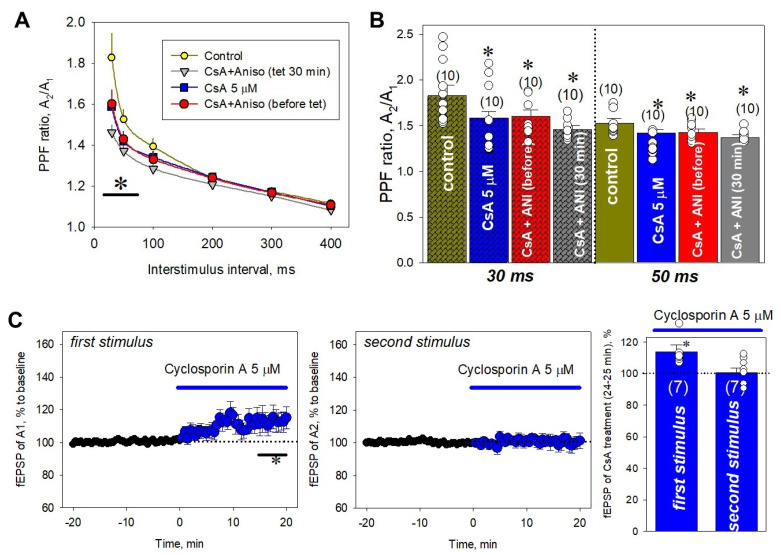
PP2B and short-term plasticity in CA1. (**A**) Cyclosporin A (5 µM) did not restore the paired-pulse facilitation ratio (PPFR) impaired by 25 µM anisomycin before tetanus (red circles) or 30 min after it (gray triangles). The application of 5 µM cyclosporin A alone (blue squares) decreased the PPFR at 30 ms and 50 ms interstimulus intervals (**B**) Summarized statistics for control (dark yellow bars), Cyclosporin A (5 µM)-treated (blue bars), Anisomycin (25 µM)-treated (red bars), and Cyclosporin A (5 µM) + Anisomycin (25 µM)-treated (gray bars) at 30 and 50 ms interstimulus intervals (ISI). CsA—Cyclosporin A, ANI—Anisomycin. (**C**) The PP2B blockade-induced PPRF impairment was associated with the increase in amplitude of the first response (left panel) rather than the decrease in amplitude of the second response (middle panel). Right panel shows averaged data for influence of cyclosporin A on the PPFR. Number of independent experiments is indicated in parentheses. * *p* ˂ 0.05 vs. the control.

**Figure 7 ijms-22-04857-f007:**
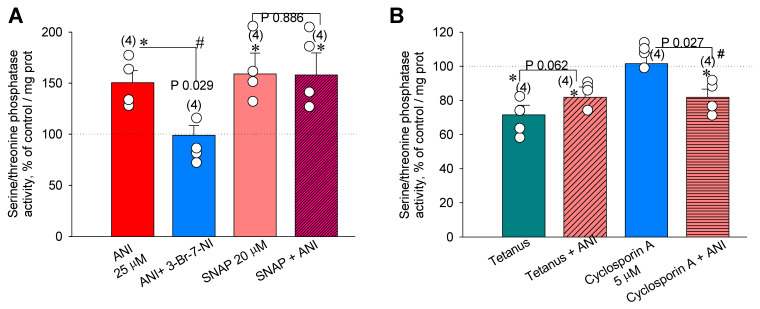
Direct measurements of serine/threonine phosphatase activity in hippocampal slices. (**A**) Application of anisomycin (25 µM, red bar) and NO donor SNAP (20 µM, pale red bar) increased the phosphatase activity in the hippocampus. A blocker of nNOS, 3-Br-7-NI (5 µM, blue bar), totally abolished the anisomycin effects on the phosphatases. Simultaneous addition of SNAP (20 µM) + Anisomycin (25 µM, violet bar) had no synergistic effect in respect to stimulation of serine/threonine phosphatase activity. (**B**) Tetanus (with four 100 Hz bursts spaced 5 min apart) drastically reduced the serine/threonine phosphatase activity (cyan bar), such that 25 µM anisomycin could not fully return it to a control level (obliquely shaded pale red bar). The PP2B blocker Cyclosporin A (5 µM) inversed the effects of anisomycin (25 µM) on the serine/threonine phosphatase activity (horizontally shaded pale red bar). The influence of Cyclosporin A (5 µM) alone to the phosphatase activity is indicated at blue bar. ANI, anisomycin. Number of independent experiments is indicated in parentheses and empty circles at the corresponding bars. * *p* ˂ 0.05 vs. the control. # *p* ˂ 0.05.

**Figure 8 ijms-22-04857-f008:**
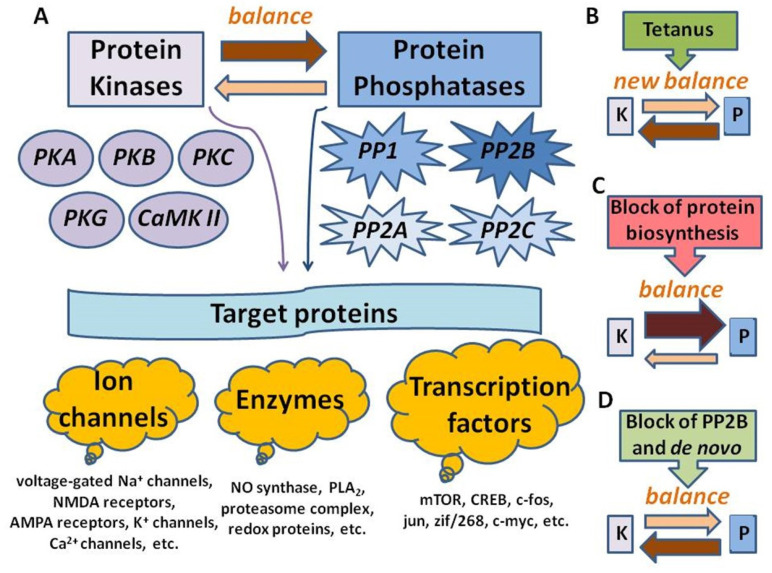
Suggested scheme of PSB-induced impairment of E-LTP. (**A**) One of the crucial mechanisms of the regulation of neuronal excitability is the phosphorylation/dephosphorylation of target proteins: ion channels, enzymes, and transcription factors. Kinase–phosphatase balance at rest is shifted towards dephosphorylation (marked by a black arrow). (**B**) Immediately after the high-frequency tetanization, the balance shifts towards kinase activity due to the induction of a wide range of intracellular protein kinases. These events lead to appearance of changes in synaptic effectivity, and activation of processes associated with protein biosynthesis. (**C**)—under the blockade of protein synthesis translation, that occurs under the PSB treatment, a kinase-phosphatase balance shifts towards the dephosphorylation processes. Ultimately, this leads to a drop in the LTP caused by tetanization. (**D**)—blockade of PP2B (calcineurin) returns the balance towards an increase in kinase activity, underlying rescue of the E-LTP, which was impaired in the PSB-treated slices. Thickness of the arrows and intensity of their shading correspond to a shift in the kinase–phosphatase balance in one direction or another. Abbreviations: PKA, cAMP-dependent protein kinase, protein kinase A; PKB, Akt kinase, protein kinase B; PKC, protein kinase C, PKG, cGMP-dependent protein kinase, protein kinase G; CaMK II, Ca^2+^ calmodulin-dependent kinase II; PP1, protein phosphatase 1; PP2A, protein phosphatase 2A; PP2B, calcineurin, protein phosphatase 2B; PP2C, protein phosphatase 2C; PLA2, phospholipase A2; mTOR, mammalian transcription factor sensitive to rapamycin; CREB, cAMP-binding element; K, the total kinase activity; P, total phosphatase activity.

## Data Availability

The data presented in this study are available on request from the corresponding author.

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
