# Peer review of "Serine/Threonine Phosphatases in LTP: Two B or Not to Be the Protein Synthesis Blocker-Induced Impairment of Early Phase"

_ijms, 2021, doi:10.3390/ijms22094857_

Round 1

Reviewer 1 Report

It is necessary to refer to figure 8, in case of being of own elaboration it must be indicated

Author Response

We thank the Editors and Reviewers for their comments that guided revision of the manuscript. Following all recommendations, we extended experimental data and clarified language throughout the text.

It is necessary to refer to figure 8, in case of being of own elaboration it must be indicated.

Response. Fig. 8 is drawn by us using standard Microsoft Office tools (PowerPoint 2007) specifically for this paper.

Reviewer 2 Report

In their study, Maltsev and colleagues investigated the contribution of calcineurin to the nNOS activation, which occurs during blockade of protein synthesis in neurons. Their experiments based on pharmacology treatment showed that the decrease of early phase LTP in the PSB-pretreated hippocampal slices are mediated by regulation of phosphatase activity and NO synthesis.

I have a major concern with the data presented in this manuscript because as stated in the introduction of the paper: « [...] early phase (E-LTP) that is protein synthesis-independent ». And the whole paper is based on the effects of protein synthesis inhibitors on this E-LTP.

Furthermore, I do not believe that understanding the molecular mechanisms responsible for the effect of protein inhibitors on LTP provides important informations to the community. Protein synthesis inhibition is a tool to differentiate two phases of plasticity but are not physiological. When you block protein synthesis for 1 hour you will completly change the homoestasis of the system and create a lot of perturbation and the conclusions that can be drawn from these experiences do not reflect what happens in a physiological situation. The study of the modulation of PP2B or NO activity during the different phases of LTP is, however, a very interesting question.

Finally, the quality of the manuscript is poor. It is very difficult to follow the reasoning of the authors and a lot of work is needed to clarify the figures and the text. Here are some examples :

- Figure 1 would be clearer if panel C took the place of panel B. It would make sense to describe the plasticity part first and then the PPF part. It is the same problem in the text : « Fig 1C shows the first response immediately after the LTP induction in control (green circles), cycloheximide- (dark red circles) and anisomycin-treated (red circles) slices, demonstrating an obvious PSB-induced E-LTP deterioration. » This part was already explain before !

- Figure 2A, 3A and C the color are poorly visible in the fEPSP time course and also in the insert.

- Title « 2.3 PP2B involvement in the PSB influences in CA1 neurons ». This title is not informative, it is vague and clearly poorly chosen

- Title « 2.4 PP2B and PSB-induced NO production in the hippocampus ». Authors never show that PP2B induced a production of NO. They showed that PP2B inhibitor block PSB-induced NO production.

-Figure 4 and 5. Why presenting control experiments (control & tetanus) after Anisomycin panel ? And in B : they put + Cyclosporin A 5 µM. But + what ? Control + CsA (because control is the last panel) or ANI + CsA ?

Other critical aspect :

in the result section : In the figure 1B, the reduction observed with ANI is the same before and after LTP induction but the authors said : « 25 μ M anisomycin decreases PPF at ISI of 50 and 100 ms » for control experiement and « PPF in anisomycin-treated slices is significantly suppressed at 30, 50 and 100 ms » for the PPF after LTP induction. This distinction is not statistically justified.

Results section 2.3 : « Cyclosporin A [...] produce any significant changes in synaptic plasticity during the LTP development not in the early neither late phases ». In the figure 3A, the late phase at 150 min is clearly affected in CsA experiment.

- Figure 4 : Nothing is explain here. How the quantify the fluorescence increase. When they measure this NO increase... And we have a clear lack of control in this figure like the effect of CsA alone or NO donor or NOS inhibitor.

The discussion should be rewritten and streamlined. It should focus on the conclusions of the experiments made. For example, I don't understand why the authors wrote a whole paragraph on NFAT and PP2B. It does not add anything to the discussion and understanding of their results.

Finally, the final figure 8 which is supposed to explain the effect of PSBs on E-LTP, manages to present about 20 partners but never NOS, NO and protein synthesis inhibitors.

Author Response

We thank the Editors and Reviewers for their comments that guided revision of the manuscript. Following all recommendations, we extended experimental data and clarified language throughout the text.

1) I have a major concern with the data presented in this manuscript because as stated in the introduction of the paper: « [...] early phase (E-LTP) that is protein synthesis-independent ». And the whole paper is based on the effects of protein synthesis inhibitors on this E-LTP. Furthermore, I do not believe that understanding the molecular mechanisms responsible for the effect of protein inhibitors on LTP provides important informations to the community. Protein synthesis inhibition is a tool to differentiate two phases of plasticity but are not physiological.

Response Absolutely correct comment. The fact that protein synthesis blockers impair both the E-LTP and PPF ratio suggest a translation-independent action of PSB that was not described before. Apparently, anisomycin and cycloheximide drive immediate signal events in the hippocampus which are not related to protein synthesis. In this regard, for example, it may be a well established anisomycin-induced stimulation of stress-activated JNK and p38 enzymes [Chung et al., 2000; Rosser et al., 2004]. Cycloheximide, beside the translation influence drives a small GTPase RhoA signaling [Darvishi and Woldemichael, 2016]. In Conclusion it is noted: “PSBs suppressed early phase of the LTP development as well as hippocampal short-term plasticity regardless of their effects on translation processes”. This seems to be a significant aspect since protein synthesis blockers are used in laboratory practice to model some memory impairments [Desgranges et al., 2008; Gonzalez-Franco et al., 2019; Kim and Cho, 2020]. Understanding the precise mechanisms of the PSB action can help to avoid misinterpretation of the experimental data.

2) - Figure 1 would be clearer if panel C took the place of panel B. It would make sense to describe the plasticity part first and then the PPF part. It is the same problem in the text : « Fig 1C shows the first response immediately after the LTP induction in control (green circles), cycloheximide- (dark red circles) and anisomycin-treated (red circles) slices, demonstrating an obvious PSB-induced E-LTP deterioration. » This part was already explain before !

Response. We swapped panels (B) and (C), and corrected the text.

3) Figure 2A, 3A and C the color are poorly visible in the fEPSP time course and also in the insert.

Response. We corrected thickness of colored lines in the Figs.

4) Title « 2.3 PP2B involvement in the PSB influences in CA1 neurons ». This title is not informative, it is vague and clearly poorly chosen.

Response. We corrected the title to: “PP2B is involved in the PSB-induced impairment of E-LTP in CA1 neurons”.

5) - Title « 2.4 PP2B and PSB-induced NO production in the hippocampus ». Authors never show that PP2B induced a production of NO. They showed that PP2B inhibitor block PSB-induced NO production.

Response. We corrected the title to: “PP2B blocker prevents the PSB-induced NO production in the hippocampus”.

6) Figure 4 and 5. Why presenting control experiments (control & tetanus) after Anisomycin panel ? And in B : they put + Cyclosporin A 5 µM. But + what ? Control + CsA (because control is the last panel) or ANI + CsA?

Response. We clarified the inscription in Fig.4B: “Anisomycin 25 µM + Cyclosporin 5 µM”. Also, the inscription in Fig. 5B: “Anisomycin 25 µM + Tetanus + Cyclosporin 5 µM”.

7) Other critical aspect: in the result section : In the figure 1B, the reduction observed with ANI is the same before and after LTP induction but the authors said : « 25 μ M anisomycin decreases PPF at ISI of 50 and 100 ms » for control experiement and « PPF in anisomycin-treated slices is significantly suppressed at 30, 50 and 100 ms » for the PPF after LTP induction. This distinction is not statistically justified.

Response. For 1C top panel (ANI before LTP induction), the p-value for the PPF interstimulus interval 30 ms was 0.058 that is not significant difference, while after LTP induction the difference was significant at all intervals. We changed the text to be more clear. 

8) Results section 2.3 : « Cyclosporin A [...] produce any significant changes in synaptic plasticity during the LTP development not in the early neither late phases ». In the figure 3A, the late phase at 150 min is clearly affected in CsA experiment.

Response. In the Fig. 3A (as well 3C) the L-LTP phase which is affected corresponds to the PSB+Cyclosporine group (3A – cycloheximide, 3C – anisomycin, respectively). Curve for the Cyclosporin A alone is at the level of control LTP curve, and is masked by it. We have reduced the thickness of the circles in the figure, and you can see that the curves are awfully close to each other.

9) Figure 4: Nothing is explain here. How the quantify the fluorescence increase. When they measure this NO increase... And we have a clear lack of control in this figure like the effect of CsA alone or NO donor or NOS inhibitor.

Response. In Material and Methods, 4.3 it is noted: “For semiquantitative estimation of the NO production was used a ratio of digitized dye signals in the layer dendrites (CA1, CA3, DG) to the DAF-FM fluorescence in the region of neuron soma in the same layer. For the precision of normalization, the area from which the average signal was calculated (region of interest, ROI) was unchanged both in the dendrites and soma layers”. In the revised version we added Suppl. Fig.3 where clearly demonstrate action of endogenous NO donor, Arginine, Ca2+-dependent stimulation of NO synthesis through NMDA application, and the prevention of anisomycin-induced NO production in the presence of NO synthase blocker, L-NNA.

10) The discussion should be rewritten and streamlined. It should focus on the conclusions of the experiments made. For example, I don't understand why the authors wrote a whole paragraph on NFAT and PP2B. It does not add anything to the discussion and understanding of their results.

Response. Some data about PP2B-NFAT signaling are presented because this pathway is involved in the neuron outgrowth which is important for the plasticity and hippocampal functioning. We simplified the Discussion, tried to focus on our results.

11) Finally, the final figure 8 which is supposed to explain the effect of PSBs on E-LTP, manages to present about 20 partners but never NOS, NO and protein synthesis inhibitors.

Response. In Fig. 8, the NO synthase is mentioned under ‘Enzymes’ group, members of which are totally regulated through phosphorylation/dephosphorylation processes. PSB influences are related to 8C and 8D panels, suggesting switch of kinase-phosphatase balance (K-P) under PSB application, and coming back of K-P during the PSB+PP2B blockade.

References

Chung KC, Kim SM, Rhang S, Lau LF, Gomes I, Ahn YS. Expression of immediate early gene pip92 during anisomycin-induced cell death is mediated by the JNK- and p38-dependent activation of Elk1. Eur J Biochem. 2000 Aug;267(15):4676-84. doi: 10.1046/j.1432-1327.2000.01517.x.
Darvishi E, Woldemichael GM. Cycloheximide Inhibits Actin Cytoskeletal Dynamics by Suppressing Signaling via RhoA. J Cell Biochem. 2016 Dec;117(12):2886-2898. doi: 10.1002/jcb.25601.
Desgranges B, Lévy F, Ferreira G. Anisomycin infusion in amygdala impairs consolidation of odor aversion memory. Brain Res. 2008 Oct 21;1236:166-75. doi: 10.1016/j.brainres.2008.07.123. 
González-Franco DA, Bello-Medina PC, Serafín N, Prado-Alcalá RA, Quirarte GL. Effects of anisomycin infusions into the dorsal striatum on memory consolidation of intense training and neurotransmitter activity. Brain Res Bull. 2019 Aug;150:250-260. doi: 10.1016/j.brainresbull.2019.06.005.
Kim WB, Cho JH. Encoding of contextual fear memory in hippocampal-amygdala circuit. Nat Commun. 2020 Mar 13;11(1):1382. doi: 10.1038/s41467-020-15121-2.
Rosser EM, Morton S, Ashton KS, Cohen P, Hulme AN. Synthetic anisomycin analogues activating the JNK/SAPK1 and p38/SAPK2 pathways. Org Biomol Chem. 2004 Jan 7;2(1):142-9. doi: 10.1039/b311242j.

Round 2

Reviewer 2 Report

The answers provided by the authors are satisfactory and the modifications of the manuscript make it more readable.